# Evaluation of Chemical and Sensory Characteristics of Sauerkraut Juice Powder and its Application in Food

**DOI:** 10.3390/foods12010019

**Published:** 2022-12-21

**Authors:** Liene Jansone, Zanda Kruma, Evita Straumite

**Affiliations:** Department of Food Technology, Faculty of Food Technology, Latvia University of Life Sciences and Technologies, Rigas Iela 22, LV-3004 Jelgava, Latvia

**Keywords:** spray-dried, fermented cabbage juice, NaCl, substitute, food application

## Abstract

Sauerkraut juice is rich in bioactive compounds; however, it is considered a byproduct of the production process. An innovative solution was found through the process of spray-drying to obtain sauerkraut juice powder. The aim of this study was to evaluate chemical and sensory characteristics of sauerkraut juice powder (SJP) and its application in foodstuffs. For SJP, total phenol content, antiradical activity, and nutritional value were determined, and the results showed that SJP is rich in minerals, especially calcium and potassium, as well as organic acids and vitamin C. SJP contains 12% NaCl and a total phenol content of 359.54 mg GAE 100 g^−1^ dw. SJP has umami attributes, such as sweet, sour, and salty. Sensory tests—descriptive, rate-all-that-apply, overall liking, and volatile profile determination—were carried out separately in SJP experimental samples with olive oil and sour cream. Among the sweet, sour, and salty flavours, garlic, yogurt, and mayonnaise were also mentioned. In the detection of volatile compounds, leafy and grassy green aromas with light almond were identified in the samples with olive oil and butter and rancid cheese and fishy/amine odours were identified in samples with sour cream. There were significant differences in the overall likability of samples, but the experimental samples with SJP were more popular than control samples; therefore, SJP may be used as a salt alternative in food applications.

## 1. Introduction

Sauerkraut juice is an agricultural byproduct that is formed in the production of fermented cabbage, and is rich in valuable bioactive compounds [1,2] such as phenolic compounds, glucosinolates and minerals [3], organic acids, sugars, biogenic amines [4], and vitamin C, which can range from 14.7 to 75 mg 100 g^−1^ in fresh weight [3]. Traditionally, sauerkraut is obtained via spontaneous fermentation induced by shredding cabbage and mixing it with salt (NaCl). In this process, cell juices are released and beneficial conditions for the development of lactic acid bacteria are created [5]. Spontaneously fermented sauerkraut contains fatty acid esters, resulting in pleasant fruity notes [6]. By the end of fermentation, vast changes occur as the cabbage becomes sauerkraut, creating unique textures and flavours, improved nutritional value with organic acids that exhibit antioxidant activity [1,4,7], increased digestibility with lactic acid bacteria as a probiotic, stabilized microflora [8,9], as well as reduced risks of fever, food poisoning, and various illnesses [2]. Additionally, scientific evidence shows that sauerkraut provides anticarcinogenic and antioxidant effects, as well as reduces inflammation and even DNA damage [10]. However, sauerkraut juice is considered a byproduct, and its applications in the food or retail industries are limited due to its short shelf life, bioactive compounds, and specific aroma. A specific sauerkraut flavour forms during fermentation and is affected by lactic acid bacteria strains and their potential to metabolize certain lipids, amino acids, glucosinolates, etc. [11]. Isothiocyanates are an important flavour-forming compound group for all *Brassica* family plants, including cabbages, and their metabolites are key odorants of sauerkraut. These compounds are also beneficial from a nutritional perspective [12].

In the current literature, to the best of our knowledge, there is no research on sauerkraut juice powder. For broader applications in the industry, spray-dried sauerkraut juice was obtained using a vertical, laboratory-scale spray dryer using starch solution as the wall material. On an industrial scale, it was obtained using a horizontal spray dryer with maltodextrin as the wall material [10].

In the food industry, NaCl is an important ingredient as it affects the physio-chemical structure of the foods [13], enhancing its palatability [14] and microbiological safety [15]. The complexity of salty flavours was first investigated in 1898, and according to early analyses by Bartoshuk [16], the taste of salt comprises different tastes of its components: Na+ stimulates taste buds, whereas Cl boosts this salty taste but has no taste of its own. Inguglia et al. [15] state that NaCl is the saltiest and purest available taste from salts [16,17]. Mostly, we perceive a salty taste via the taste buds on our tongue in the oral mucosa [17].

However, a high NaCl intake (more than 5 g a day) can increase the risk of cardiovascular and other diseases, according to recommendations in both the USA and Europe [18,19]. The most common yet discreet way to exceed daily salt intake is by consuming processed foods, such as sausages, cheese, breads, smoked meats, snacks, etc. [20]. Salad dressings are also a hidden source of NaCl [21]. In commercially available salad dressings with olive oil, the amount of salt ranges from 0.0 to 5.5 g 100 g^−1^ and in yogurt/mayonnaise type garlic sauces, it is 0.9–3.8 g 100 g^−1^, based on the information available on the home pages of various producers and wholesalers.

There have been investigations of various methods of salt reduction in food applications [14,17,22,23], such as substitutes, replacers or enhancers [18,20], reduction of particle size, encapsulation, etc.

The most common salt (NaCl) replacer is potassium chloride (KCl) [20,24], but calcium chloride (CaCl2) and magnesium chloride (MgCl2) have also been tested [15]. Though this is not the case for all food applications, K and Ca can create an off-flavour or bitter taste and not meet the required processing and quality parameters achieved by NaCl. A different method to reduce salt is to encapsulate it so that it dissolves only in the mouth, using saliva to ensure the correct sensory profile [23]. Salt, encapsulated in fat, releases its taste only in the mouth by inhomogeneous distribution, creating ‘salty spots’ described as taste contrast technology; therefore, these samples are characterised as being saltier [21,25,26].

Another salt substitute is notoriously known monosodium glutamate (MSG) [15], delivering an umami taste that can “modulate sweet, enhance salty and suppress bitter taste” [26] and make reduced-salt food palatable with a more pleasant taste, rich intensity, and softer mouthfeel. Umami attributes naturally occur in some traditional foods, for example, fermented animal- or plant-based products [25,26]. SJ powder could be applied as a pure salt substitute, being a healthy alternative that still provides a salty taste with considerable amounts of minerals, vitamins, and phenols. Similarly, dried seaweed is also used in food applications as a salt substitute due to its natural saltiness [27,28]. The potential of sauerkraut juice powder in food and cosmetics is currently under study, but its limitations are its high content of NaCl and specific flavour characteristics.

An innovative product, sauerkraut juice powder, was developed, and the aim of this study was to evaluate the chemical and sensory characteristics of sauerkraut juice powder (SJP) and its applications in foodstuffs.

## 2. Materials and Methods

The experiment was carried out at Latvia University of Life Sciences and Technologies, Faculty of Food Technology, Department of Food Technology.

### 2.1. Sauerkraut Juice Powder

Sauerkraut juice was collected and delivered in 1 L bottles by Ltd. “Dimdiņi” (Latvia), a company that ferments cabbage using traditional technology. Cabbages of different varieties were shredded, and 1.7% NaCl (“Artiomsol”, Ukraine), 1% grated carrot, and 0.2% caraway seeds were added. All ingredients were mixed and pressed to encourage anaerobic conditions and left to ferment for 14 days. In the fermentation process, cell juices called sauerkraut juices were released, and constituted 30% of the total cabbage weight. In the sauerkraut packaging process, the juice was removed and regarded as an industrial byproduct. On average, sauerkraut juice contained soluble solids °Brix 7.6–8.5 and a pH of 3.7.

The delivered sauerkraut juice was spray-dried on a vertical, laboratory-scale spray dryer (Buchi 290) using starch solution as the wall material [29]. Starch (pure, soluble starch, 162.10 g mol^−1^, CHEMPUR) was heated (30 min., 90 °C) in deionized water at a ratio of 1:20. The solution was then cooled overnight and mixed with sauerkraut juice, at a ratio of 1: 0.75. The mixture was stirred on a magnetic stirrer and immediately spray dried to obtain sauerkraut juice powder (SJP). It was then stored in two Ziplock bags at room temperature until further analyses.

### 2.2. Commercially Available Salad Dressings and Garlic Sauces

Samples of commercially available salad dressings and garlic sauces were chosen due to the popularity and taste preferences of the region, and according to their prices and availability in local markets. Locally available olive oil salad dressings were as follows: Oak’a Burger garden salad dressing (Dormers Ltd., Latvia), Oak’a Burger balsamic salad dressing (Dormers Ltd., Latvia), Kraft balsamic vinaigrette (Kraft Heinz Foods Company, CA, USA), Spilva vegetable dressing (Orkla Foods Latvia Ltd., Rīga, Latvia), and Spilva lemon/olive oil dressing (Orkla Foods Latvia Ltd., Rīga, Latvia). Garlic sauces were as follows: Heinz (Kraft Heinz Foods Company, CA, USA), Spilva (Orkla Foods Latvia Ltd., Rīga, Latvia), Hellman’s (Unilver, Englewood Cliffs, NJ, USA), Taste Me (Sanitex Ltd., Rīga, Latvia), and Kitchen masters (UAB Kedainiu Konservu Fabrikas, Kedaini, Lithuania). The nutritional value, composition, sugar, and salt content of these dressings and sauces were compared.

### 2.3. Preparation of the Experimental Samples

To prepare samples for the experiment, a fine salt “Artiomsol” (Ukraine), extra virgin olive oil “Fratelli Mantova” (Italy), and 20% fat content sour cream “Rimi” (Latvia) were used. The samples were prepared and marked as shown in Table 1.

The salt content of the experimental samples was chosen based on market research on similar, commercially available products, described in Section 2.2 and Section 3.2. Based on these findings, a maximum amount of 2.0 g 100 g^−1^ NaCl (OOC; OO1) was used in the experimental samples with olive oil, and then the amount was reduced to 1.5 (OO2) and 1 (OO3) g 100 g^−1^. A maximum amount of 0.8 g 100 g^−1^ NaCl (SCC; SC1) was used in sour cream samples and reduced to 0.5 (SC2) and 0.2 (SC3) g 100 g^−1^. The equivalent salt content was then calculated for sauerkraut juice powder; to obtain 2 g of salt in the experimental sample, it was multiplied by 100 g and divided by 12 g (12 g of salt in 100 g of sauerkraut juice powder). Then, the amount of salt was reduced accordingly in the following samples.

### 2.4. Analytical Methods

#### 2.4.1. Nutritional Value of Sauerkraut Juice Powder

The nutritional value and mineral content of sauerkraut juice powder was determined in collaboration with laboratory group Ltd. J.S. Hamilton Baltic, Latvia, accredited and certified by international standards and requirements.

The energy value was determined according to Regulation (EU) No. 1169/2011, and mass spectrometry was used to determine the mineral profile by ionization with inductively coupled plasma (ICP-MS), determined according to PB-223/ICP, ed. II. Dietary fibre was determined according to AOAC 991.43:1994; ash content according to PN-A-75101-08:1990 + Az 1:2002; sodium according to PB-318/FAAS, ed. I of 27.07.2015,; sodium chloride was calculated as Na × 2.5; and vitamin C according to PB-135/HPLC, ed. II of 15.09.2015. For the sugar profile, enzymatic-spectrophotometry was used. Carbohydrates were calculated as dietary fibre and total sugar content. The moisture content was determined according to ISO 939:2021, which is an oven-drying method, at 105 ± 1 °C.

Total phenol concentration was determined using the Folin–Ciocalteu reagent according to Singleton et al. [30] with slight modifications, as described in previous studies, as well as the antiradical activity caused by a DPPH assay [31].

#### 2.4.2. Sensory Evaluation of SJP Application in Food with Olive Oil and Sour Cream

Experimental SJP application samples with the sensory taste and aftertaste of both olive oil and sour cream was developed and assessed by a group of 10 experts [32]. Experts were asked to write down in rough notes any words associated with the taste, aftertaste, and sensation intensity of the experimental samples. The experts developed a vocabulary from 38 terms for taste and aftertaste, which were modified and reduced to 16 for samples with olive oil and 22 for samples with sour cream, based on expert group discussions.

From the list of developed vocabulary of taste and aftertaste, terms included in the RATA test (rate-all-that-apply) were sweet, salty, sour, bitter, spicy, garlic, cabbage, yogurt, mayonnaise, cottage cheese, and vinegar. Thirty untrained panelists, aged 18–64, recruited based on their willingness to try the samples, were asked to check the taste and aftertaste attributes to characterize experimental samples and rate the intensity of each taste attribute using a five-point structured scale (1—none to 5—very intense) [33,34]. The sensory evaluation took place in a specially designed room with suitable lightning in a noise-free and well-ventilated environment. Each of the samples was served in a transparent plastic container (30 mL) marked with a three-digit number, and water and crisp bread pieces were provided as palate cleansers between samples.

The percentage of female participants was 80%, and 83% were aged 18–25 years old.

#### 2.4.3. Determination of Volatile Compounds

To determine the volatile compounds, a solid-phase microextraction (SPME) technique was used. Then, 0.5 g of sauerkraut juice powder and 5 g of each experimental sample were weighed in a glass vial, stirred, and heated at 35 °C for 10 min to equilibrate headspace and 30 min with Carboxen™/Polydimethylsiloxane (CAR/PDMS) fibre (Supelco Inc., Bellefonte, PA, USA) to extract volatiles. Volatile compounds were analysed by GC/MS, as described by Galoburda et al. [35]. Compounds were identified using the Nist database.

### 2.5. Statistical Analysis

The results are expressed as the mean value ± standard deviation. Significant differences (*p* ≤ 0.05) among the acquired samples were determined by a *t*-test. Analysis of variance (ANOVA) was used to evaluate the results of overall likability.

## 3. Results

### 3.1. Nutritional Value of Sauerkraut Juice Powder

In the current experiment, vertically spray-dried sauerkraut juice was tested, and its characteristics and detailed nutritional value are shown in Table 2. Its energy values and carbohydrates were mostly acquired by the addition of starch solution to sauerkraut juice to enable spray drying and included sugars such as glucose, fructose, and maltose. In the scientific literature, there is a lack of studies on spray-dried vegetable-based products. For a comparison in basic nutritional value, naturally salty dried seaweed was chosen and is shown in Table 2. The seaweed samples were either air-dried or freeze-dried and then milled into a powder form. The parameters of seaweed fall within a wide range, mainly influenced by the variety of seaweed. There are brown, green, and red species of seaweed or algae, and each of these contains different bioactive compounds that contribute to their rich and unique nutritional values [27]. Seaweeds are used in food applications for their thickening properties as well as their potential for fat and salt reduction, but they often suffer from a decline in sensory characteristics [27,28].

The ash content in SJP is 15.28 g 100 g^−1^, and thus, there is a wide array of mineral nutrients, as shown in Table 3. The NaCl content contributes to the amount of ash, as shown in Table 2. SJP may not be considered an everyday condiment, yet the amount of minerals present in the powder is significant. However, Vilar et al. [28] concluded that the ash content in seaweed is higher than that of land plants, and seaweed is rich in Na, K, and Fe.

Daily reference intakes for vitamins and minerals, according to EU regulation 1169/2011, are presented in Table 4. A product is considered to be significantly nutritious if it contains at least 15% of nutrient reference values per 100 g. Potassium being the most abundant element in SJP, contributed most to daily intake, along with manganese, calcium, and iron. SJP contained a significant amount of iron, which met the recommended daily intake.

The spray-drying process for vitamin C was gentle, and fairly large amounts remained in SJP. In this specific sample, the amount of vitamin C was 103 mg 100 g^−1^, though its content was affected by many obstacles associated with agriculture, processing, and storage.

### 3.2. Commercial Samples of Salad Dressings and Garlic Sauces

For the evaluation of sauerkraut juice powder in food applications, the composition of commercial sauces was first investigated to estimate the ingredients and salt and sugar content in the samples, as shown in Figure 1. The estimation of salt content in commercial salad dressings and garlic sauces was necessary to determine the amount of SJP (and calculate its salt equivalent) necessary for the experimental samples in this study. The sugar content was estimated for informational purposes as it naturally occurs in the SJP, whereas it is added in the production of commercial dressings and sauces.

Five oil-based salad dressings and five cream-based garlic sauces, available in the local markets, were analysed in this study. The salt content in the garlic sauces ranged from 1.3 to 3.8 g 100 g^−1^, with a median of 1.9 g, and the difference among the samples was 2.5 g. Salt content in the salad dressings was similar, ranging from 1.2 to 5.5 g 100 ^−1^, with the median being 1.8 g, and the difference among the samples was 4.3 g. In order to visualize and compare the salt amount in all of the studied samples, a percentage of daily intake [18,19] was calculated in a portion-size serving. The weight of a portion-size serving was determined based on single-serving condiment packets used by catering and fast food businesses, and the results are compiled in Table 4.

All of the sauces exceeded 5% of the recommended daily salt intake, two of the sauces exceeded 10%, and one sauce, Ltd. Spilva salad dressing, even exceeded 27%. This is a considerable amount of salt that greatly contributes to the daily intake and may be considered a hidden source of NaCl.

Sugar content in the salad dressings was significantly higher than in the garlic sauces, ranging from 8 to 17 g 100 g^−1^, with an equal median and average of 13. The sugar content in the garlic sauces ranged from 5.2 to 8.7 g 100 g^−1^, with an average and median of 7.

The common components in the salad dressings were water, vinegar, olive oil (also soybean or canola oil), sugar, salt, spices, stabilizers, and preservatives. In commercially available cream-based garlic sauces, the following were found in various amounts: water, sugar, rapeseed oil (also sunflower oil), modified maize or starch, egg yolk powder with maltodextrin, milk/whey/yogurt powder, garlic (different amounts of added garlic 0.5, 0.7, 5%), garlic–chive mixture (11%), parsley, mustard, chives, vinegar (citric, malic, ascorbic, or lactic acid), thickeners (xanthan or guar gum), preservatives (potassium sorbate, sorbic acid, or sodium benzoate), flavourings, and antioxidants. There was great variation in salt content and energy value in the analysed commercial samples, as shown in Table 4.

For the experimental samples, pure olive oil and sour cream were used; vinaigrettes were avoided due to the varied effects of the different ingredients and their interactions with SJP. The experimental samples in the current study were only prepared using salt (control) and SJP, with no addition of sugar, garlic, yogurt, mayonnaise, vinegar, or other ingredients.

The minimum amount of salt in the estimated commercially available garlic sauces was 1.3 g 100 g^−1^. However, due to previous sensory tests in an expert group (not described in this study), experimental samples prepared with SJP with an equivalent salt amount of 1.3 g were defined as too salty; hence, 0.8 g 100 g^−1^ was used.

### 3.3. Sensory Evaluation of Experimental Samples with Olive Oil and Sour Cream

#### 3.3.1. Frequency of Use of Sensory Terms

Experimental samples with olive oil.

In the control sample (OOC), predominantly sweet, sour, and salty tastes were identified, as shown in Table 5, whereas in the SJP samples, sour and salty tastes were predominant. Most of the participants identified the control sample as sweet, more so than the experimental samples; yet, the sweet aftertaste lingered longer in samples OO1 and OO3. Sample OO2 was recognized as having the sourest taste and aftertaste. Salty taste was identified the least in the control sample with just salt and olive oil, whereas in the experimental samples with SJP, a salty taste was identified by 93–100% of participants. Even OO3, the sample with the least salt content, was identified as having a salty taste more often than the OOC. Additionally, the salty aftertaste in the DSJP samples was identified by more participants. This salty aftertaste and mouthfeel can be explained by the encapsulation effect of oil, which allows salty particles to linger in the mouth for longer and dissolve slower.

In sample OO1, with an SJP salt equivalent of 2%, the term “garlic” was used by 73% of participants, with 47% identifying a garlic aftertaste. The garlic taste in the OO2 and OO3 samples was also recognized by 50 and 37% of participants, respectively, with a lingering aftertaste. A cabbage taste was identified in the OO1 and OO2 samples by more than 40% of the participants.

There were no significant differences in bitter and spicy tastes and aftertastes. In the expert group discussion after the taste characterization descriptive test, bitter and spicy tastes were attributed more to the peculiarity of the taste of extra virgin olive oil.

Experimental samples with sour cream

Sour and salty tastes in the sour cream samples, as shown Table 6, were mentioned the most (by more than 60% of participants in all the samples). The salty taste was equally identified for the control sample (SCC) and experimental sample with equivalent salt in the SJP (SC1). A garlic taste was identified by more than 60% of participants for the samples SC1 and SC2, and more than 40% in SC3. A yogurt taste was mentioned in all of the samples by more than 50% of participants, and more than 57% detected a mayonnaise taste in DSJP samples SC1 and SC2. Tastes such as cottage cheese, vinegar, cabbage, spicy, and bitter were identified in the sour cream samples by less than 50% of participants.

In the control sample, sour, salty, sweet, and yogurt tastes were identified the most, (by more than 50% of participants). In the samples with SJP, the same tastes, as well as garlic, mayonnaise, and yogurt tastes, were also identified, although no such ingredients were added. The identified aftertastes were similar: sour and salty with yogurt, mayonnaise, and cottage cheese tastes.

Sour cream samples had a sour aftertaste in all of the samples. SC1 and SC2 had a salty aftertaste and the control sample SCC had a sweet aftertaste.

#### 3.3.2. Sensory Attributes of the Sample

The total rated points (RATA from 1 to 5 points) in the samples regarding sensory attributes are compared in a radar chart.

Experimental samples with olive oil

The intensity of the 11 taste attributes in samples with olive oil are shown in Figure 2. Five tastes had significant differences—salty, sour, spicy, garlic, and vinegar—with the greatest differences in samples with the highest amount of SJP and salt equivalents of 2 and 1.5% (OO1 and OO2). Additionally, differences in sweet and garlic tastes were mentioned, as well as cabbage and spicy tastes.

In the samples with olive oil, the intensity of the aftertaste was significantly reduced. In the control sample (OOC), the bitter taste and aftertaste was mentioned the most. This may be connected to the somewhat bitter olive oil aftertaste. The aftertaste included salty, cabbage, and vinegar tastes.

Experimental samples with sour cream

Regarding the intensity of the samples with sour cream, a variety of tastes was mentioned. Panelists marked all 11 tastes, albeit with significant differences, as shown in Figure 3. The intensity of the taste attributes in SC1 was significantly different compared with the other samples. The dominating tastes of the SC1 sample (salt equivalent of the SCC sample) were sweet, sour, salty, garlic, and mayonnaise. Additionally, the aftertaste remained intense. There were no significant differences between the SCC and SC2 samples (reduced amount of salt), yet garlic and mayonnaise tastes prevailed in SC2. In SC3 (SJP salt equivalent of 0.2), the intensity of the yogurt taste and aftertaste was rated the highest.

The intensity of the aftertastes of the sour cream samples were significantly reduced, yet the variety of rated tastes increased. All taste attributes were mentioned, with the most dominant being salty, sour, garlic, and mayonnaise. Mayonnaise and yogurt were mentioned just as often as cabbage, garlic, and cottage cheese aftertastes.

#### 3.3.3. Overall Liking of the Samples

There were significant (*p* ≤ 0.05) differences among the samples when rating overall likability, either in oil or sour cream samples, as shown in Figure 4. The liking of these samples was ranked from 1–5: 1—do not particularly like; 5—like the most.

The overall liking of olive oil samples, OO1 and OO2, with an SJP salt equivalent of 2 and 1.5% were liked the most, and the sample with a reduced amount of salt (OO3—1.0% salt equivalent) was liked more than the control sample, but with no significant differences.

There were significant (*p* ≤ 0.05) differences between the sour cream samples. For SC1, there was no rating of 5 (like the most), and in total, among the samples with the SJP, most participants liked these samples the least, which may be explained by their strong taste intensity. The sample SC3 (reduced amount of salt with least SJP) was as liked as SC1. This could be explained by quite the opposite reason, due to not enough taste intensity. However, the average liking was rated more than 3 in both samples, compared with SCC. SC2, with a 0.5% salt equivalent, was liked the most, being described as not too salty and with the most taste variation. Despite the differences in the SJP salt equivalent, all of the samples were liked more than the SCC, leading to the conclusion that sauerkraut juice powder could be used as an alternative to salt.

#### 3.3.4. Volatile Compounds in the Experimental Samples

The sensory evaluation of experimental samples with sauerkraut juice powder brought out vast and unusual taste attributes. Due to these findings, volatile compounds were tested to compare subjective sensory tests to the objective micro-extraction of volatiles.

There were 11 volatile compounds found in the experimental samples with olive oil, as shown in Table 7. The addition of SJP increased acid compound content. Overall, peak areas in the control sample were higher than in the samples with SJP. Most of the determined aldehydes and alcohols found in our control sample were characteristic of olive oil from different regions of Turkey [38].

Not all of the volatile compounds were found in every sample. Some were present in the samples with the smallest amount of added SJP, but not shown in the control sample. Meanwhile, the volatile compound peak was present in the control sample but disappeared in the SJP samples. In total, all of the samples had a leafy, grassy green aroma with a distinct sourness and slight almond aroma.

Similar results were acquired in the sour cream samples, where 17 volatile compounds were detected (Table 8); however, not all of the compounds were represented in all of the samples.

The volatile compounds found in each of the samples were acids, representing sweet, sour, and cheese-like aromas. A buttery odour was detected in the control sample (SCC) and SC1, whereas a fishy/amine odour was detected in the two samples with the highest SJP additions (SC3, SC2). Most of the detected volatile compounds were characteristic of sour cream, as investigated by Shepard et al., [39] where sour, sweet, and somewhat salty taste attributes were identified. No garlic odour was identified in any of the experimental samples.

## 4. Discussion

In order to exploit byproducts, sauerkraut juice was vertically spray-dried, acquiring sauerkraut juice powder and using starch solution as the wall material. The obtained sauerkraut juice powder (SJP) had salty, sweet, and sour taste attributes, reminiscent of an umami taste. It also contained a significant amount of minerals, such as potassium, calcium, magnesium, and iron, as well as vitamin C, which could be considered for daily reference intakes. The obtained sauerkraut juice powder contained both Ca and K trace elements. Potassium and magnesium are the most common sodium substitutes; however, they have bitter nuances in taste when substituting for salt [13,24]. Additionally, K and Ca have been used and explored as Na substitutes [14,15].

Salt is used in traditionally fermented sauerkraut, and it remains in acquired products, such as sauerkraut juice powder. In this study, SJP, as a salt alternative, was liked in all of experimental samples. Moreover, samples with a reduced amount of salt (equivalent) were equally liked. Additionally, SJP provided samples with extra flavours, such as garlic and mayonnaise, and with sweet and sour tastes. As Taladrid et al. [40] concluded, additional tastes provoke sensations and mask the absence of NaCl, creating an enhanced flavour

To compare nutritional values, dried seaweed was explored due to its natural saltiness. Seaweed in food applications is used for several reasons, including thickening and as a substitute for fat and salt [27,28,37]. However, as with sauerkraut juice powder, the amount of seaweed used in food preparation is limited because of its specific sensory attributes [28]

Sensory evaluation, as a quality parameter, is very important in cabbage and sauerkraut products due to their distinct aroma. Different SJP applications were trialled in foods such as meat and bread, as well as experimental samples with olive oil and sour cream. Sample tests were chosen according to the most popular oil-based and cream-based salad dressings.

Neither NaCl nor sauerkraut juice powder dissolve in fat/oil, providing an inhomogeneous, ‘salty spot’ taste contrast [17,20,21]. Commonly, a salty sensation in the mouth is quickly adjusted to and becomes addictive [25]. The encapsulation of salt can deliver that salty mouthfeel without increasing the salt content of a product [23]. In the samples with olive oil, salt and sauerkraut juice powder are somewhat encapsulated in fat and provide a salty taste and mouthfeel, while adding extra flavour.

A sensory evaluation of the experimental samples with olive oil found that sweet, sour, salty, and bitter tastes were rated the highest, whereas volatile compounds detected a leafy, grassy green odour and sourness, almond, and burned sugar aromas.

Both salt and SJP dissolve in sour cream, and sauerkraut juice powder brought out sweet, salty, and sour taste attributes, creating an umami taste and mouthfeel [26]. The sensory evaluation marked sweet, sour, salty, and yogurt tastes in all of the samples. Mayonnaise and garlic flavours were mentioned for SC1 and SC2 (highest SJP amounts).

The volatile compounds in all of the sour cream samples brought out sweet, sour, rancid, and cheese aromas. A fishy/amine aroma was detected in SC1 and SC2, whereas a buttery odour was detected in SCC and SC3 samples.

The taste intensity in the experimental sour cream samples was more pronounced than in the experimental olive oil samples.

In all of the samples, cabbage taste or aroma was not significantly mentioned or identified.

## 5. Conclusions

SJP contains minerals, vitamins, phenols, sugars, and NaCl. The sensory evaluation marked the preference of olive oil and sour cream samples with the addition of SJP. The volatile compounds in olive oil samples had a leafy and grassy green aroma with a distinct sourness and slight almond aroma, characteristic of olive oil. The volatile compounds in sour cream samples were characteristic of sour cream, with fishy/amine odours being detected in the SJP samples. Sauerkraut juice powder could be used as a salt alternative in food applications if it is acceptable in taste references.

## Figures and Tables

**Figure 1 foods-12-00019-f001:**
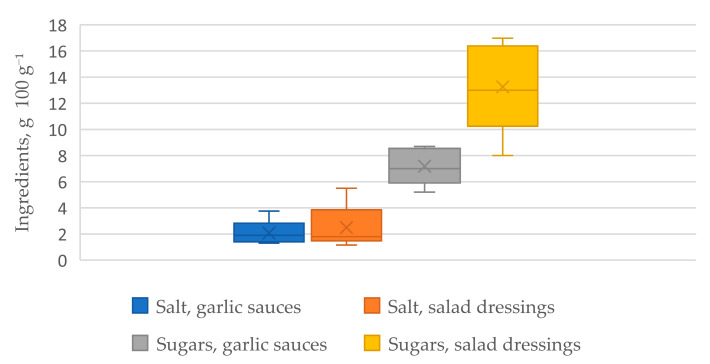
Average salt and sugar content in commercially available salad dressings and garlic sauces.

**Figure 2 foods-12-00019-f002:**
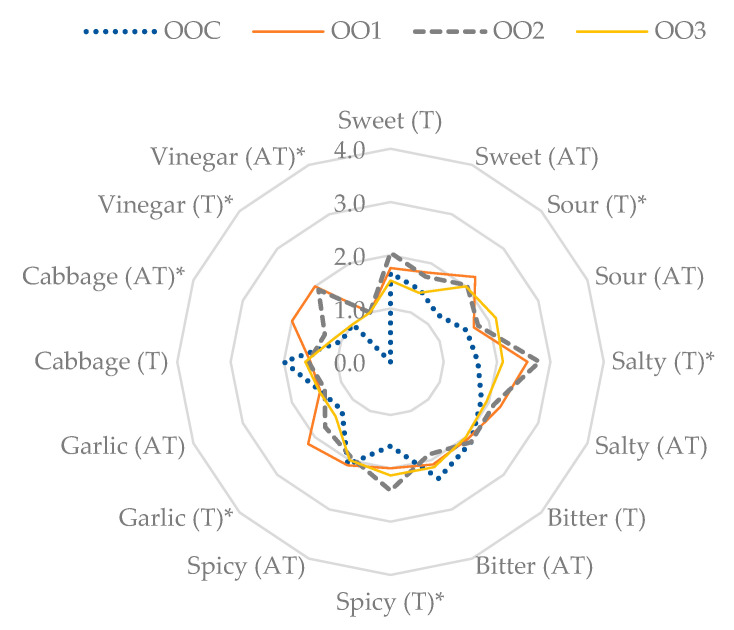
Sensory attributes of taste (T) and aftertaste (AT) of salt and equivalent salt of sauerkraut juice powder in samples with olive oil. * These tastes and aftertastes represent significant differences (*p* ≤ 0.05).

**Figure 3 foods-12-00019-f003:**
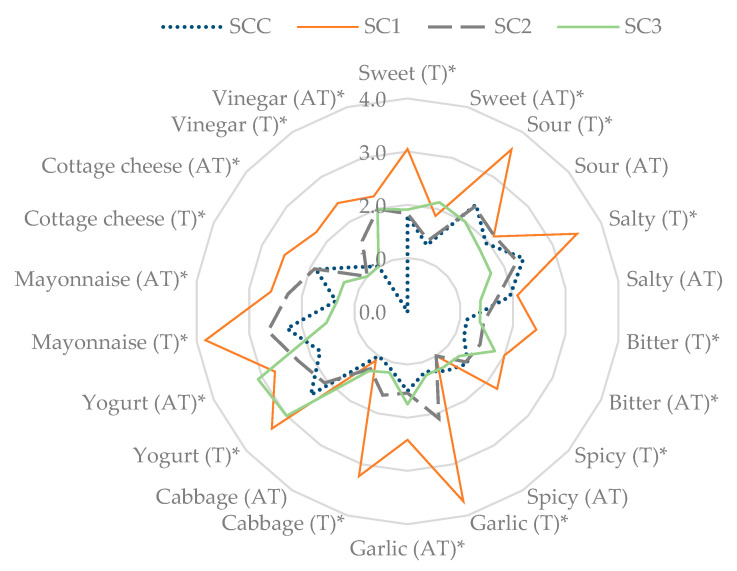
Sensory attributes of taste (T) and aftertaste (AT) of the salt and equivalent salt of sauerkraut juice powder in the samples with sour cream. * These tastes and aftertastes represent significant differences (*p* ≤ 0.05).

**Figure 4 foods-12-00019-f004:**
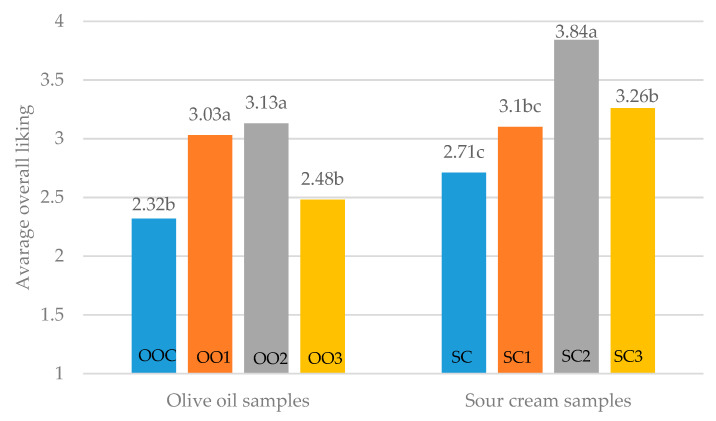
Overall liking of the olive oil and sour cream samples. Different letters among the samples mark significant differences (*p* ≤ 0.05).

**Table 1 foods-12-00019-t001:** Samples and abbreviations of NaCl, DSJ salt equivalent, and DSJ amounts used in the samples.

Type of Dressing	Abbreviation	NaCl, g 100 g^−1^	SJP Salt Equivalent, g 100 g^−1^	SJP, g 100 g^−1^
Experimental samples with olive oil	OOC	2.0	-	-
	OO1	-	2.0	16.7
	OO2	-	1.5	12.5
	OO3	-	1.0	8.3
Experimental samples with sour cream	SC	0.8	-	-
	SC1	-	0.8	6.7
	SC2	-	0.5	4.2
	SC3	-	0.2	1.7

**Table 2 foods-12-00019-t002:** The physico-chemical and nutritional value of vertically spray-dried sauerkraut juice and dried seaweed 100 g^−1^.

Parameter	SJP, Vertical	Seaweed, Different Varieties (from—to)	Reference
Moisture, %	11.82 ± 0.33	8.99–16.31	[28]
NaCl, %,	12.05 ± 0.16	5.88–36.77	[28]
Total phenol content, mg GAE, 100 g^−1^ dw	359.54 ± 18.26	0.86–328.7	[27]
Antiradical activity by DPPH, mg TE 100 g^−1^ dw	22.62 ± 0.71	-	
**Nutrition value 100 g^−1^**			
Energy value, kcal	294.7	11.88	[36]
Carbohydrates	61.8		
—Including sugars		<1	
Glucose	8.7	-	
Fructose	4.3	-	
Maltose	9.8	-	
Total sugars	27.1	-	
—Dietary fibre	5.4	2.5–9.5–15.7	[36,37]
Protein	6.5	6.3–27.2	[27,37]
Fat	<0.1	2	
Ash	15.28	17.50–44.03	[27,28,36]

**Table 3 foods-12-00019-t003:** Vitamin C and minerals in SJP and daily reference intake.

Minerals (mg)	mg 100 g ^−1^	DRI *	15% of the Nutrient Reference Values
Magnesium	93	375	56.2
Manganese	11	2	0.3
Coper	1	1	0.15
Potassium	1534	2000	300
Calcium	312	800	120
Iron	15	14	2.1
Vitamin C	103	80	12

* DRI—daily reference intake

**Table 4 foods-12-00019-t004:** Salt content and energy value in commercially available salad dressings and garlic sauces.

With Olive Oil	NaCl, g 100 g^−1^	Per Serving, 25 g	% Daily Intake (5 g)	Energy Value, kcal
Kraft balsamic vinaigrette	1.2	0.3	6.0	219
Oak’a Burger, garden salad dressing	1.8	0.5	9.0	167
Oak’a Burger, balsamic dressing	1.8	0.5	9.0	182
Spilva salad dressing	5.5	1.4	27.5	30
Spilva lemon/olive oil dressing	2.6	0.7	13.0	50
OOC	2.0	0.5	10.0	807
OO1	2.0	0.5	10.0	812
OO2	1.5	0.4	7.5	816
OO3	1.0	0.3	5.0	817
With sour cream or mayonnaise				
Heinz garlic	1.3	0.3	6.5	371
Hellmann’s garlic	1.5	0.4	7.5	283
Taste Me garlic	1.9	0.5	9.5	405
Kitchen masters garlic sauce	1.9	0.5	9.5	405
Spilva garlic sauce	3.8	0.9	18.8	285
SCC	0.8	0.2	4.0	198
SC1	0.8	0.2	4.0	200
SC2	0.5	0.1	2.5	200
SC3	0.2	0.1	1.0	200

**Table 5 foods-12-00019-t005:** The percentage of terms used to describe the taste (T) and aftertaste (AT) of the experimental samples with olive oil.

Samples %	Sweet	Sour	Salty	Bitter	Spicy	Garlic	Cabbage
	T/AT	T/AT	T/AT	T/AT	T/AT	T/AT	T/AT
OOC	80/43	80/37	83/43	73/63	40/37	23/13	20/3
OO1	70/53	93/43	100/73	67/77	53/63	73/47	47/30
OO2	67/37	100/60	97/67	73/57	40/53	50/30	43/30
OO3	63/50	77/47	93/67	67/47	50/43	37/30	17/23

**Table 6 foods-12-00019-t006:** The percentage of terms used to describe the taste (T) and aftertaste (AT) of the experimental samples with sour cream.

Samples, %	Sweet	Sour	Salty	Bitter	Spicy	Garlic	Cabbage	Yogurt	Mayonnaise	Cottage Cheese	Vinegar
	T/AT	T/AT	T/AT	T/AT	T/AT	T/AT	T/AT	T/AT	T/AT	T/AT	T/AT
SCC	50/47	87/67	93/57	27/20	33/30	37/20	27/10	50/33	47/47	30/50	23/0
SC1	67/50	87/70	93/80	30/18	43/27	60/40	43/33	63/50	60/57	43/50	40/13
SC2	60/43	80/63	77/67	30/19	37/23	67/43	47/23	57/40	57/50	40/30	47/7
SC3	73/50	77/63	60/43	27/18	47/27	40/27	33/20	60/40	43/30	43/33	23/3

**Table 7 foods-12-00019-t007:** The percentage of volatile compound peak areas in the experimental samples with olive oil.

Volatile Compound	RT *	OOC	OO1	OO2	OO3	Sensory Characteristics
Aldehydes						
2-Hexanal	18.26	12.28	3.69	5.69	4.30	Leaf, green
Benzaldehyde	26.11	6.49	0.44	1.85	0.82	Almond, burned sugar
Esters						
3 Hexen-1ol, acetate (E)	20.87	9.88	4.83	6.93	4.50	Sharp fruity green aroma reminiscent of unripe pear or banana
Alcohols						
Hexen-1ol	22.49	6.77	2.90	3.83	1.73	Grassy green aroma
3,7,11 Trimethyl-1-dodecanol,	30.42	-	-	-	3.21	
benzyl alcohol	32.13	17.20	0.73	2.34	1.66	Sweet, flower
4-Methyl-2,4-bis (4trimethylsilyloxyphenyl) pentene 1	25.66	11.03	-	-	-	
Acids						
Acetic acid	24.05	18.51	84.37	75.19	50.14	Sour
Pentanoic acid	31.51	2.70	0.56	0.39	-	Sweet
N-decanoic acid	38.48	1.26	0.87	0.85	3.64	Rancid, fat

* RT—retention time.

**Table 8 foods-12-00019-t008:** The percentage of volatile compound peak areas in the experimental samples with sour cream.

Volatile Compound	RT *	SCC	SC1	SC2	SC3	Sensory Characteristics
Aldehydes/Ketones						
1,1-Dimethyl hydrazine,	24.13	-	33.44	-	27.92	Ammonia-like
1,4-Difluoro 1,3-butadiyne,	10.49	--	3.86	7.26	4.19	
Esters						
3-Hydroxy2-butanone	20.38	5.26	-	-	2.54	Buttery odour
1,3-Dioxol-2-one	10.46	7.99	-	-	-	
Alcohols						
Benzyl alcohol	32.16	-	-	1.36	0.34	Sweet, flower
1,2-Diphenyl-1,2-ethanediol	32.15	-	1.63	-	-	
Acids						
1-Naphthalenecarboxylic acid, 8-bromo	14.5	-	-	1.63	-	Strong lemon odour
Acetic acid	24.13	42.98	33.44	44.33	27.92	Sour
Butanoic acid	27.92	18.07	12.26	18.64	12.70	Rancid, cheese, sweet
Hexanoic acid	31.52	14.15	10.01	18.38	12.91	Sour, fatty, sweet cheese
Octanoic acid	34.58	5.98	1.50	2.36	1.58	Sweet, cheese
N-decanoic acid	38.5	-	1.49	-	1.05	Rancid, fat
Nonanoic acid	38.49	-	-	1.94	-	Waxy, dirty cheese, cultured dairy
Heterocyclic Compounds						
Methylarsine dibromide	15.22	-	-	-	8.85	Pungent, acid-like
Morpholine	20.29	-	2.37	4.09	-	Fishy, amine
2-(2-Propenyl)-furan	32.15	5.57	-	-	-	

* RT—retention time.

## Data Availability

Data are contained within the article.

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
