# Peer review of "Evaluation of Chemical and Sensory Characteristics of Sauerkraut Juice Powder and its Application in Food"

_foods, 2022, doi:10.3390/foods12010019_

Round 1

Reviewer 1 Report

This study is very interesting due to the potential of SJP in the food market as a salt alternative.

The introduction is well-documented, as well as the objectives of this work. The novelty is given in terms of the SJP, accompanied by the NaCl issue, the spray-drying methodology, and the sensory evaluation.

The several experimental apparatus are the standard ones and appropriate for the study.

Results are clearly presented and explained. The discussion should be extended and more detailed.

Some references need to be checked/corrected (years of publication, etc).

Additional comments:

1 – The health implications of SJ could be mentioned in the Introduction section, as the ingredient has been used in some countries for medicinal purposes.

2 – Did the authors perform spray-drying at other temperatures? How the temperature, in this methodology, affects the quality parameters of the SJP and the flavor profiles? Was the density of SJP measured? And what about the powder recovery?

 3 – As laboratory-scale spray-drying brings some limitations (when compared to the industrial scale), the authors could discuss the yield issue, and the size range of the produced particles of SJP.

4 – Laboratory group J.C. Hamiltons needs to be properly addressed.

 5 – How was the recruitment of the untrained panelists done? Were data on body weight and on consuming frequency (of SJ) of untrained panelists considered?

6 – In the Discussion section, the authors seem to consider the results of this sensory evaluation as definitive/sufficient. Is it so? It would be interesting to compare the RATA methodology with the generic Descriptive Analysis, in terms of discriminative ability, for example.

Author Response

Results are clearly presented and explained. The discussion should be extended and more detailed. - Thank you, it has been corrected.

Some references need to be checked/corrected (years of publication, etc). - Thank you, it has been corrected.

Additional comments:

1 – The health implications of SJ could be mentioned in the Introduction section, as the ingredient has been used in some countries for medicinal purposes. - Thank you, it has been corrected.

2 – Did the authors perform spray-drying at other temperatures? How the temperature, in this methodology, affects the quality parameters of the SJP and the flavor profiles? Was the density of SJP measured? And what about the powder recovery? – There were numerous experiments carried out to spray-dry sauerkraut juice on a vertical, laboratory-scale spray-dryer since the juice is a challenging liquid, rich in sugars and organic acids. Different wall materials, their concentrations, temperature, flow, and pressure settings were explored. The settings to obtain the powder, described in this study were with the best resulting outcome.

 3 – As laboratory-scale spray-drying brings some limitations (when compared to the industrial scale), the authors could discuss the yield issue, and the size range of the produced particles of SJP. – Thank you for your remark! Since this was not really relevant to the current study it is not included. The data is in process.

4 – Laboratory group J.C. Hamiltons needs to be properly addressed. - Thank you, it has been corrected.

 5 – How was the recruitment of the untrained panelists done? Were data on body weight and on consuming frequency (of SJ) of untrained panelists considered? – Thank you! The panelists were recruited by their willingness to try the samples and the body weight and consuming frequency was not considered.

6 – In the Discussion section, the authors seem to consider the results of this sensory evaluation as definitive/sufficient. Is it so? It would be interesting to compare the RATA methodology with the generic Descriptive Analysis, in terms of discriminative ability, for example. –Yes, thank you for the suggestion and we do agree that there are various methods and will be taken into consideration in future studies.

Reviewer 2 Report

Dear authors, 

After reading the manuscript "Evaluation of sauerkraut juice powder in food applications, its  nutritional value and assessment of taste attributes ", I realized that the manuscript showed in some parts the scientific rigour wanted, but in other parts I have missed it.

The authors have presented critical evaluation only in some paragraphs.

The references are not exactly current, besides the objective could be more attractive and cientific.

Thats why I have written some suggestions below in an attempt to improve the paper.

L.33- "improved nutritional value" - How does it happen ?

L.34- " Applications of juice is limited." I didn't understand. Then, detail the limitations. Would that be a possible rationale for your paper ? So, place it in a most appropriate paragraph.

L.43- 47 - I think you could organize the paragraphs more effectively. Perhaps, improving on this approach... it got confused - all mixed:  Nutrition, sensory and compounds. You can improve and become it more organized.

L.37 and L 40 - I think it was early to mention . "powder" at this point. Line 53 seems more adequate to me. It is possible that I may have felt a confusing approach in L. 43-47 because of this.

L.54 - "health beneficial properties" please, point out which ones.

L. 61- guglia et al. - ??

L.82 - If sauerkraut juice powder has been tested as a salt substitute it seems relevant to me to review the title and the objectives ( your abstract as well) within this line of action. At least that's what I was led to believe by reading several paragraphs - L.56 - L.86.

Suggestion : Evaluation of chemical and sensory characteristics of sauerkraut juice powder as a salt alternative in food ( sauce and dressing) application - You did not evaluate only "nutritional", and you did not evaluate only "taste". Perhaps, focusing more on "sauce" and "dressing" would be more coherent.

L.94- 2.1- Did you follow any author?

L.95- How were the samples homogenized?

L. 96 - "cabbage of different varieties" - I did not understand.

L.151- only Flavor ??

Has the project been submitted to an evaluation by a university ethics committee? Did it follow the Helsinki Declaration? Please, enter the approval protocol number.

 How many sessions were conducted ? What attributes were evaluated ?

L.168- Give more details about the participation of the assessors.

 L.173- Check authors, please.  

L.194- Moisture as a nutritional ? I couldn't find in your Materials and methods how it was analyzed.  

L.199- E. Garicano Vilar et al - Please correct according to Journal's rules.

L.208 -  I did not find Minerals in your Material and Methods. Please, correct.

L. 352- Standardize the scales in Figure 4 and 5, use the same maximum value for both (3,5) . Maybe, even merge them into the same graph.

361- The scale of the "Avarage overall liking" in Figure 5 could be 1.0 in 1.0 or 1.5 in 1.5.  ( See previous suggestion)

L.367 - Check number of  this subtitle, since this analysis is part of sensory 

L.377 and L.390 - Sensory characteristics instead of organoleptic, please.

It would not be possible to merge the results of all the tables and graphs, for example table 7 and table 8. I missed the fact that I could compare more easily.

L.431- The conclusion needs to answer your objectives. The conclusion was attached to the discussion and only about the sensory, it needs to pay attention to the chemical part as well.  

Best wishes

Author Response

Thank you for your time and input in our study! We really appreciate your suggestions and comments!

The references are not exactly current– Thank you, it has been corrected.

 besides the objective could be more attractive and cientific.- Thank you for your suggestion, it has been corrected.

L.33- "improved nutritional value" - How does it happen ? - Thank you, it has been corrected.

L.34- " Applications of juice is limited." I didn't understand. Then, detail the limitations. Would that be a possible rationale for your paper ? So, place it in a most appropriate paragraph. – The use of sauerkraut juice in its raw state is limited, as only it is offered as a drink, pasteurized, packed in bottles, but the rest of the leftover juice needs to be utilized.

L.43- 47 - I think you could organize the paragraphs more effectively. Perhaps, improving on this approach... it got confused - all mixed:  Nutrition, sensory and compounds. You can improve and become it more organized. – Thank you, it has been corrected.

L.37 and L 40 - I think it was early to mention . "powder" at this point. Line 53 seems more adequate to me. It is possible that I may have felt a confusing approach in L. 43-47 because of this. -– Thank you, it has been corrected.

L.54 - "health beneficial properties" please, point out which ones. -  Thank you, it has been corrected.

  1. 61- guglia et al. - Thank you, it has been corrected.

L.82 - If sauerkraut juice powder has been tested as a salt substitute it seems relevant to me to review the title and the objectives ( your abstract as well) within this line of action. At least that's what I was led to believe by reading several paragraphs - L.56 - L.86. – Thank you, it has been corrected.

Suggestion : Evaluation of chemical and sensory characteristics of sauerkraut juice powder as a salt alternative in food ( sauce and dressing) application - You did not evaluate only "nutritional", and you did not evaluate only "taste". Perhaps, focusing more on "sauce" and "dressing" would be more coherent. – Thank you.We intentionaly did not use the terms dressings and sauces, since those are finished products, whereas our experimental samples were not such, but with a potential in perspective.

L.94- 2.1- Did you follow any author? – The study was carried out within the EU project in collaboration with the production farm Ltd “Dimdiņi” and as a partner, they delivered us the raw material – sauerkraut juice.

L.95- How were the samples homogenized? – The samples were not homogenized. During spray drying, the mixture of starch solution and sauerkraut juice was constantly stirred on a magnetic stirrer.

  1. 96 - "cabbage of different varieties" - I did not understand. – There are many studies on fermenting cabbage, its physico-chemical changes, comparing certain varieties. As well as we have compared several varieties. But in this study juice was collected from fermented cabbage of mixed varieties. According to the company’s recipe to reach the quality of sauerkraut, the influence of season, specific variety composition of carbohydrates etc., is taken into consideration, therefore sauerkraut is made mixing several varieties.

L.151- only Flavor ?? - According to ISO 5492 flavor is a complex combination of the olfactory, gustatory and trigeminal sensation perceived during tasting. When creating the vocabulary of sensory properties, the focus was on the attributes of taste and aftertaste, which are included in the definition of "flavor". This paragraph has been clarified.

Has the project been submitted to an evaluation by a university ethics committee? Did it follow the Helsinki Declaration? Please, enter the approval protocol number. The present study is not considered to be a health research study, therefore it can be conducted without approval from the committees.

 How many sessions were conducted ? What attributes were evaluated ? – There were three sessions conducted in the expert group and the evaluation with the untrained panelists was conducted once. Attributes are describet in chaper 2.4.2.

L.168- Give more details about the participation of the assessors. - Thank you, it has been corrected.

 L.173- Check authors, please. - Thank you, it has been corrected.

L.194- Moisture as a nutritional ? I couldn't find in your Materials and methods how it was analyzed.  - Thank you, it has been corrected.

L.199- E. Garicano Vilar et al - Please correct according to Journal's rules. - Thank you, it has been corrected.

L.208 -  I did not find Minerals in your Material and Methods. Please, correct. - Thank you, it has been corrected.

  1. 352- Standardize the scales in Figure 4 and 5, use the same maximum value for both (3,5) . Maybe, even merge them into the same graph. - Thank you, it has been corrected.

361- The scale of the "Avarage overall liking" in Figure 5 could be 1.0 in 1.0 or 1.5 in 1.5.  ( See previous suggestion) - Thank you, it has been corrected.

L.367 - Check number of  this subtitle, since this analysis is part of sensory - Thank you, it has been corrected.

L.377 and L.390 - Sensory characteristics instead of organoleptic, please. - Thank you, it has been corrected.

It would not be possible to merge the results of all the tables and graphs, for example table 7 and table 8. I missed the fact that I could compare more easily. – Thank you, we do agree that visual comparison could be more clear with the merged tables, but yes, in this case it would gaine the opposite effect.

L.431- The conclusion needs to answer your objectives. The conclusion was attached to the discussion and only about the sensory, it needs to pay attention to the chemical part as well - Thank you, it has been corrected.

Round 2

Reviewer 2 Report

After another evaluation of the manuscript, I see some improvement in the quality of the paper. The authors have accepted some of my requests, and those they did not accept, they have justified..

English is always useful to ask a native speaker for a final appreciation.

They added more authors to better substantiate the discussion and improved objectives and conclusion.

Author Response

Dear reviewer,

Thank you, again, for your time and input in our study! Your comments and suggestions have been really valuable and helpful.

Best of wishes,

Liene Jansone